# Solubility Improvement of Progesterone from Solid Dispersions Prepared by Solvent Evaporation and Co-milling

**DOI:** 10.3390/polym12040854

**Published:** 2020-04-07

**Authors:** Xing Chen, Ioannis Partheniadis, Ioannis Nikolakakis, Hisham Al-Obaidi

**Affiliations:** 1The School of Pharmacy, University of Reading, Reading RG6 6AD, UK; cpuchenxing@hotmail.com; 2Department of Pharmaceutical Technology, School of Pharmacy, Faculty of Health Sciences, Aristotle University of Thessaloniki, 54124 Thessaloniki, Greece; ioanpart@pharm.auth.gr

**Keywords:** solid dispersions, mechano-chemical activation, saturation solubility, thermal analysis, Flory–Huggins, Hansen solubility parameters

## Abstract

The aim of this contribution was to evaluate the impact of processing methods and polymeric carriers on the physicochemical properties of solid dispersions of the poorly soluble drug progesterone (PG). Five polymers: hydroxypropyl methylcellulose (HPMC), hydroxypropyl methylcellulose acetate succinate (HPMCAS), microcrystalline cellulose (MCC), polyvinylpyrrolidone (PVP) and silica (SiO_2_), and two processing methods: solvent evaporation (SE) and mechano-chemical activation by co-milling (BM) were applied. H-bonding was demonstrated by FTIR spectra as clear shifting of drug peaks at 1707 cm^−1^ (C20 carbonyl) and 1668 cm^−1^ (C3 carbonyl). Additionally, spectroscopic and thermal analysis revealed the presence of unstable PG II polymorphic form and a second heating DSC cycle, the presence of another polymorph possibly assigned to form III, but their influence on drug solubility was not apparent. Except for PG–MCC, solid dispersions improved drug solubility compared to physical mixtures. For SE dispersions, an inverse relationship was found between drug water solubility and drug–polymer Hansen solubility parameter difference (Δδt), whereas for BM dispersions, the solubility was influenced by both the intermolecular interactions and the polymer T_g_. Solubility improvement with SE was demonstrated for all except PG–MCC dispersions, whereas improvement with BM was demonstrated by the PG–HPMC, PG–PVP and PG–HPMCAS dispersions, the last showing impressive increase from 34.21 to 82.13 μg/mL. The extensive H-bonding between PG and HPMCAS was proved by FTIR analysis of the dispersion in the liquid state. In conclusion, although SE improved drug solubility, BM gave more than twice greater improvement. This indicates that directly operating intermolecular forces are more efficient than the solvent mediated.

## 1. Introduction

Poor aqueous solubility has always been a challenge in the development of hydrophobic drugs for oral delivery [1,2]. One potential solution has been the use of solid dispersions in which the hydrophobic drug is mixed at molecular level with a hydrophilic carrier. Molecular interactions such as hydrogen bonding can be optimized by careful choice of the preparation conditions [3]. While enhancement in solubility could be attributed to transformation of the drug to amorphous state, the nature of carrier was also shown to have a major impact on dissolution [1,4]. Although nowadays state of the art mechnanochemical methods such as hot-melt extrusion are exploited for their ability to enhance drug release in conjunction with polymers, there are only a few reports for their application to progesterone, e.g., reservoir systems of progesterone loaded into polymeric matrix for controlled release [5]. Mechano-chemical activation is often applied by dry or wet milling and is used to enhance the solubility and dissolution rate of poorly soluble drugs [3,6]. This is usually applied by co-milling drug with excipients and is commonly used for particle size reduction. It has the advantage of avoiding solvent use, thus simplifying the process compared with solvent-based methods such as spray drying or freeze-drying [4]. 

The main drawback of mechano-chemical activation is the heterogeneous nature of collisions between drug and excipients during co-milling, which may lead to large batch-to-batch variations [7]. While a number of studies have been conducted on aqueous solubility using mechano-chemical activation, the issue of variability in the resulting drug solubilities has been overlooked [8]. An important question is what is the impact of the processing method (co-milling vs. solvent evaporation) on molecular interactions. The answer to this question will aid a better understanding of both these processes, optimization of the applied conditions and minimization of batch-to-batch variation. In general, differences are expected between the two methods since mechano-chemical activation involves heat/collisions, whereas solvent evaporation involves solvent-mediated interactions. The use of these methods is common for preparation of solid dispersions. PG solid dispersions with Pluronic, polyethylene glycol and polyvinylpyrrolidone were previously prepared via solvent evaporation and were found to improve solubility and vaginal permeability [9]. In a different study, PG solid dispersion with Pluronic 127 were formulated as mucoadhesive vaginal tablets and were found to improve vaginal drug delivery with minimum side effects [10].

In this study, solid dispersions of progesterone (PG) used as a poorly water-soluble model drug were prepared by solvent evaporation and mechano-chemical activation. PG is an endogenous steroid hormone involved in the menstrual cycle, pregnancy and embryogenesis [11]. Oral administration of PG has proven ineffective due to its poor aqueous solubility, 7 µg/mL at 25 °C, and extensive first-pass metabolism [12,13]. Despite attempts by formulation scientists to improve its solubility via micronization or oil-based dispersions [14,15,16,17], these formulations could only achieve a bioavailability of <10% owing to PG’s first-pass metabolism [18]. Thus, large doses, typically, 100, 200 and 300 mg, are required, which have been associated with low tolerance in patients. Therefore, improvement of PG solubility was attempted by co-processing with different polymeric excipients and application of solvent evaporation and co-milling.

Five excipients were compared: hydroxypropyl methylcellulose (HPMC), hydroxypropyl methylcellulose acetate succinate (HPMCAS), microcrystalline cellulose (MCC), polyvinylpyrrolidone (PVP) and silica (SiO_2_). They were selected on the basis of their different molecular characteristics: HPMC, HPMCAS and PVP are amorphous but PVP does not have H-bond donor groups, whereas the other two polymers have, MCC is semicrystalline with H-bond donors, and silica is amorphous with an ability to accept protons and form H-bonds [19]. In addition, these polymers are commonly used excipients for oral drug delivery, and enhancement of aqueous solubility based on these polymers will be of great interest to pharmaceutical formulation scientists. The hydrophilic nature of those polymers dictates their ability to increase dissolution and wettability of hydrophobic drugs. Drug–excipient interactions in the solid dispersions were elucidated experimentally using thermal and spectroscopic analysis and also theoretically, using Hansen solubility parameters and Flory–Huggins analysis. Parameters of these models were calculated as predictors of the miscibility of progesterone with the examined polymers (Figure 1).

## 2. Materials and Methods

### 2.1. Materials

Progesterone (PG) was obtained from Sigma Aldrich (Dorset, UK). Hydroxypropyl methylcellulose acetate succinate (HPMCAS grade HF, average *M*_W_ 20,000 Da) was gifted by Shin-Etsu (Tokyo, Japan). Polyvinylpyrrolidone (PVP K30, average *M*_W_ 40,000 Da), silicon dioxide (SiO_2_), hydroxypropyl methylcellulose (hypromellose, HPMC) (viscosity range of 3000–5600), microcrystalline cellulose (MCC) and deuterium oxide (D_2_O) were obtained from Sigma Aldrich (Gillingham, UK). All other solvents were obtained from VWR International (Leicestershire, UK). All chemicals were of analytical grade and were used as received. Chemical structures of the experimental drug and polymers are presented in Figure 1.

### 2.2. Methods

#### 2.2.1. Preparation of PG Solid Dispersions Using Solvent Evaporation

Mixtures of PG with each polymeric excipient were prepared using 1 g drug/polymer samples at ratio 3:1. Due to limited solubility of some of the polymers, higher polymer/drug ratios required high solvent volumes making the process unpractical. The samples were dissolved in 50 mL acetone under agitation for 20 min and then transferred to a round-bottom flask. Acetone was removed using a Rotavapor (BUCHI Rotavapor^®^ R-300, Flawil, Switzerland) at 40 °C. After evaporation, the solids retained in the flask were dried under vacuum for further 24 h to eliminate any residual solvent. The resultant solid mass was ground for 10 min and passed through 80 μm sieve.

#### 2.2.2. Preparation of PG Solid Dispersions Using Mechano-chemical Activation

Mixtures of PG with each polymeric excipient were prepared using 1 g samples at drug/polymer weight ratio of 1:3 which was previously found optimal for enhancing stability of amorphous PG [20]. The mixtures were transferred into 10 mL zirconium oxide lined grinding jars containing one 12 mm diameter aluminium oxide ceramic milling ball and milled at a frequency of 30 s^−1^ for 5 min using a Retsch MM200 mill (Castleford, UK). The product was collected and sieved with an 80 μm orifice sieve.

#### 2.2.3. Preparation of Physical Mixtures

Physical mixtures of PG with each polymer at ratios corresponding to those used in the PG–polymer dispersions were prepared by mixing for 10 min in a Turbula mixer (Bachofen, Switzerland).

#### 2.2.4. Differential Scanning Calorimetry (DSC)

Samples of 5–10 mg were placed in sealed aluminium pans and scanned using a differential scanning calorimeter (DSC) (TA instruments, New Castle, DE, USA) at a heating rate of 10 °C/min, from 25 °C to 230 °C. The method started with isothermal hold at 90 °C for 10 min. Melting points and heats of fusion were determined. 

From the heat of fusion of the pure drug (Δ*H*_ο_), considered as 100%, and the heat of fusion of the drug from dispersion (Δ*H*_f_), the crystallinity (*X*c) was calculated according to Equation (1) for SE dispersions and Equation (2) for BM dispersions.
*X*_c_ (%) = [(Δ*H*_f_)/(0.75)]/Δ*H*_ο_)) × 100(1)
*X*_c_ (%) = [(Δ*H*_f_)/(0.25)]/Δ*H*_ο_)) × 100(2)

#### 2.2.5. Fourier-Transform Infrared Spectroscopy (FTIR) for Solid Dispersions

Solid dispersions and physical mixtures were analysed by FTIR spectroscopy to identify hydrogen bonding. Spectra were collected using a Perkin–Elmer Spectrum One (Waltham, MA, USA) spectrophotometer, operated in attenuated total reflectance mode. Samples were placed on germanium crystal and scanned over the wavelength range of 4000–550 cm^−1^ at a resolution of 4 cm^−1^.

#### 2.2.6. FTIR Studies of Solid Dispersions Dissolved in D_2_O and Methanol

In certain cases, FTIR spectra of solid dispersions were also obtained after dissolving the dispersions in methanol and D_2_O in order to confirm the existence of H-bonding between drug and polymer in the absence of water peaks. A Perkin–Elmer Spectrum One spectrometer (Waltham, MA, USA) was used, and spectra of the dissolved samples were collected with the aid of an Omni infrared cell with calcium fluoride windows (Sigma Aldrich, Dorset, UK). Scans were collected over the wavelength range of 4000–1100 cm^−1^ at a resolution of 4 cm^−1^.

#### 2.2.7. Solubility Studies

The equilibrium solubility of PG from solid dispersions and physical mixtures in phosphate buffer was determined. Solubility of neat PG was also determined for comparison. Samples containing the equivalent of 10 mg PG were placed in microcentrifuge tubes containing 1 mL phosphate buffer and mixed using a rotary mixer (Bibby Scientific Limited, Staffordshire, UK) at room temperature (~22 °C) for 72 h. The drug-saturated samples were subsequently centrifuged (Heraeus Biofuge Pico, Germany) at 13,000 rpm (16,060× *g*) for 10 min after which the supernatant was removed and analysed. The dissolved drug was quantified by UV spectrophotometry using a standard calibration curve measured for absorbance at 245 nm. All measurements were performed in triplicate, and mean and standard deviation were calculated.

#### 2.2.8. Drug–Polymer Miscibility

##### Solubility Parameters

The total solubility parameter, *δ*_t_, represents the total attractive forces in a condensed matter and can be expressed as the square root of the sum of Hansen solubility parameters (HSPs)
*δ*t = (*δ*_d_^2^ + *δ*_p_^2^ + *δ*_hb_^2^)^1/2^(3)
where *δ*_d_ accounts for the dispersion forces, *δ*_p_ for the polar forces, and *δ*_hb_ for hydrogen-bonding attraction [21]. HSPs for progesterone and HPMCAS were calculated according to the group contribution method of [22] using the following equations applicable to values greater than 3 MPa^(1/2)^:*δ*_d_ = (*∑*i*N*i*C*i + 959.11)^0.4126^(4)
*δ*_p_ = *∑*i*N*i*C*i + 7.6(5)
*δ*_hb_ = *∑*i*N*i*C*i + 7.7(6)
where, *C*i is the contribution of chemical group i occurring *N*i times in the molecule and *δ*_hb_ includes all forms of electron exchange.

For progesterone, the first-order contributing groups are: CH_3_–, –CH_2_–, –CH=, >C<, >C=O and –CH=C<–COOH and the second order group is C_cyclic_=O (Figure 1). For HPMCAS, the first-order contributing groups are: –CHO–, –CH<, –CH_2_–, –OH, CH_3_COO–, –CH_2_COO–, –CH_2_-, –COOH, –OCH_3_, –CH_3_, –CH<, OH and –CH_2_O (Figure 1). Occurrences are shown in Table 1. For HPMCAS, they were calculated by taking into account the substitutions in the glucopyranose unit which according to the manufacturer’s data (HPMCAS grade HF) were for the hydroxy (OH) group 50%, the acetyl (–OCH_2_COCH_3_) group 12%, the succinyl (–OCOCH_2_CH_2_COOH) group 6%, the methoxy (–OCH_3_) group 24% and the hydroxy proxy (–OCH_2_CH(OH)CH_3_) group 8%. Since there are six hydroxyls per glucopyranose unit, half of which is available for esterification (manufacturer’s data), three of HPMCAS –OH are esterified and three are not. Therefore, the occurrences for the substituting groups were for acetyl group 0.36 (= 12% × 3), for succinyl group 0.18 (= 6% × 3), for methoxy group 0.72 (= 24% × 3) and for hydroxy proxy group 0.24 (= 8% × 3). Group occurrences and computations of Hansen solubility parameters for PG and HPMCAS are given in Table 1. Calculated solubility parameters for PG and HPMCAS and literature values for HPMC, MCC, PVP and SiO_2_ [23,24] are given in Table 2.

##### Flory–Huggins Thermodynamic Analysis

For the determination of enthalpy, entropy and energy of mixing, the Flory–Huggins model was applied [20,25,26]. Δ*G*_M_ is the free energy of mixing for *n*_drug_ and *n*_polyner_ moles, respectively, present at *Φ*_drug_ and *Φ*p_olymer_ volume fractions. The interaction parameter *χ* accounts for the enthalpy of mixing. It can be calculated from Equation (8) and then substituted into Equation (7) to find the free energy of mixing [20].
(7)$$\frac{\Delta G_{M}}{RT}=n_{drug}ln\Phi_{drug}+n_{polymer}ln\Phi_{polymer}+\chi n_{drug}\Phi_{polymer}$$
(8)$$\left(\frac{1}{T_{M}^{\mathrm{mix}}}-\frac{1}{T_{M}^{\mathrm{pure}}}\right)=\frac{-R}{\Delta H_{\mathrm{fus}}}[ln\Phi_{\mathrm{drug}}+\left(1-\frac{1}{m}\right)\Phi_{\mathrm{polymer}}+\chi \Phi_{\mathrm{polymer}}{}^{2}$$
where $T_{M}^{mix}$ and $T_{M}^{pure}$ are the melting temperatures of the drug in the presence of polymer and alone, respectively, Δ*H*_fus_ is the enthalpy of mixing of pure drug, and *m* is the ratio of the polymer to drug volume (calculated as molar volumes from the true density).

## 3. Results and Discussion

From the chemical structures shown in Figure 1, the ability for H-bonding between drug and polymers can be deduced. HPMC and HPMCAS form H-bonds by offering protons and have been found to produce solid dispersions with improved drug solubility [19]. MCC has H-bond donors and has been found to increase ibuprofen solubility and dissolution by co-milling with the drug [27]. PVP has proton accepting carbonyl group and less strong accepting N atom and can act both as solubilizer and crystallization inhibitor [28]. Silica can accept protons to form H-bonds and has been found to improve the solubility and dissolution of drugs by co-milling, the extent of which depends on whether dry or wet co-milling is applied [29].

### 3.1. Hansen Solubility Parameters

Hansen solubility parameters (HSPs) were calculated as an indication of miscibility of the solid dispersion components. The total solubility parameter, *δ_t_*, is a measure of the attractive intermolecular forces in the material. Materials with similar *δ_t_* values are considered miscible or that one can dissolve in the other. Therefore, comparison of *δ_t_* values can be used to predict miscibility of PG with the examined polymers [30]. From the values given in Table 2, it can be seen that the contribution of dispersive forces (*δ*_d_) is similar for all materials, but the polar (*δ*_p_) and H-bonding (*δ*_hb_) components are different, with cellulose-based excipients showing higher *δ*_p_, *δ*_hb_ and hence, higher *δ*_t_ values (from 30.6 to 39.3 MPa^1/2^ compared with 18.9 to 21.9 MPa^1/2^). Consequently, the differences in *Δδ*_t_ between drug and polymers varied and were low for PVP and SiO_2_ (0.5 and 3.0 MPa^0.5^, respectively), intermediate for the cellulose ethers HPMC and HPMCCAS (11.7 and 13.8 MPa^0.5^, respectively), but high for MCC (20.4 MPa^0.5^).

Overall, the results of HSPs point to good drug miscibility with PVP and SiO_2_, low miscibility with the cellulose ethers and no miscibility with MCC. These results should reflect in the intermolecular interactions during preparation of solid dispersions by solvent evaporation, since both drug and polymer are in a dissolved state, free to interact. However, in the case of mechano-chemical activation, interactions are also governed by the degree of surface melting and creation of molten surface areas which is a prerequisite for the operation of intermolecular forces.

Hansen solubility parameters were computed in this work: for progesterone and HPMCAS according to Stefanis and Panayiotou 2008 [22]; for HPMC and MCC from Archer 1992 [24]; for PVP from Kolter et al. 2012 [31] and for SiO_2_ from Tsutsumi et al. 2019 [23].

*T*_g_ (glass transition temperature) was computed: for progesterone from Koutsamanis et al., 2020 [5]; for HPMC from Shin-Etsu Technical information manual; for HPMCAS and PVP from Lehmkemper et al. 2018 [32] and for MCC from Szczesniak et al. 2008 [33].

### 3.2. Spectroscopic Analysis of PG Solid Dispersions

#### 3.2.1. Drug–Polymer Interactions

Figure 2 presents FTIR spectra of PG and PG solid dispersions prepared by solvent evaporation (SE) and co-milling (BM). PG shows two characteristic peaks at 1707 cm^−1^ (first from the left) and 1668 cm^−1^ (second from the left) corresponding to stretching vibration of carbonyl groups at C20 and C3, respectively (Figure 1). The C3 carbonyl is highly basic and can form more than two H-bonds because of adjacent conjugation conferring the oxygen a negative charge as well as the available space for additional bond [34]. Therefore, the C3 carbonyl has major role in intermolecular H-bonding. This also depends on the presence of excess hydrogen bond donors, as it is the case with HPMCAS and HPMC.

As it can be seen in Figure 2, overall, the peaks of PG obtained from both SE and BM dispersions are shifted to lower wavenumbers (lower frequencies), which is attributed to H-bonding. The PG–HPMC dispersion from SE shows clear shift of the 1668 cm^−1^ peak (C3 carbonyl) to 1663 cm^−1^ but only a minimal shift of the 1707 cm^−1^ peak to 1705 cm^−1^ appearing as shoulder, implying that at C20, the molecule remains as unbounded ketone. Conversely, the PG–HPMC dispersion from BM shows clear shifts at both peak positions (from 1707 to 1699 and from 1668 cm^−1^ to 1662 cm^−1^), indicating greater H-bonding interaction. For the PG–HPMCAS and PG–MCC dispersions prepared by both SE and BM methods, shifts appear in both C3 and C20 groups. A broad vibration at 1737 cm^−1^ could be seen due to the C=O group of HPMCAS-PG prepared by BM, whereas it slightly shifted to 1739 cm^−1^ in HPMCAS-PG prepared by SE.

In the case of PG–PVP and PG–SiO_2_ dispersions and for both SE and BM methods, the spectra show broad, shifted peaks. PVP and SiO_2_ are hygroscopic and contain up to 10% *w*/*w* water under normal environmental conditions [35]. Since they contain only proton-accepting groups, direct H-bonding is not possible. Therefore, the shift in the PG–PVP dispersions should be attributed to H-bonding with water molecules acting as mediators between PVP and drug, whereas for PG–SiO_2_, it should be attributed to SiOH…H_2_O bonding due to the association of water with surface silanol groups [36]. With a potential coordination number of C3 carbonyl up to three, water molecules may be attracted and form bonds at C3 and to some extent, at C20.

Overall, the results of spectroscopic analysis demonstrate the ability of all the polymers to interact with PG via H-bonding. The clear peak shifts of the PG–HPMC co-milled dispersion compared to the corresponding SE at both carbonyl peak positions indicate greater H-bonding ability with possible consequences of the presence of polymer on the wettability, solubility and dissolution of the drug.

#### 3.2.2. Polymorphism

PG has been known to exist in two polymorphic forms, I and II, that can be distinguished from each other on the basis of their thermodynamic properties and FTIR spectra [37]. Since differences in the physical stability and dissolution profile of the polymorphs has been reported [38], any polymorphic transformation of PG due to co-processing is of importance. In Figure 3, FTIR spectra of solid dispersions prepared by SE (a) and BM (b) are presented. In the BM spectra of HPMC, PVP and SiO_2_, a second peak appears at 863 cm^−1^ (indicated with arrows) immediately after 871 cm^−1^ (out of plane bending to an *sp^2^* at C2-C4), which is characteristic of PG form II [35]. This peak is not seen in the corresponding SE solid dispersions which show only the characteristic peak of PG form I at 871 cm^−1^, and also, it is not seen in any SE dispersion of the other polymers. On the basis of shifting and absorption intensity of the peaks in Figure 3b, it appears that form II is predominant in the BM dispersions of PG with HPMC, form I is predominant in the BM dispersions of PG with HPMCAS and MCC, whereas both polymorphs, I and II, are present in equal amounts in the BM dispersions of PG with PVP and SiO_2_. In all the SE dispersions, only form I is present.

### 3.3. Thermal Analysis of PG Solid Dispersions

#### 3.3.1. DSC of PG Solid Dispersions Prepared by Solvent Evaporation

Thermal analysis was used to confirm the two polymorphic forms seen in the FTIR spectra and detect further thermal events. In Figure 4, thermograms of PG–polymer solid dispersions prepared by SE are shown (thermograph for neat drug with single peak at 131 °C is not shown to avoid confusion). In Table 3, results of melting points, heats of fusion and heats of crystallization after second heating cycle are presented. Since PG exhibits polymorphism, it is not surprising to see in Figure 4 more than one endothermic peak evolving during heating instead of the single peak of neat drug at 131 °C (not shown in figure due to crowdedness). Furthermore, it is known that the melting points of the two well-known PG polymorphs, I and II, are 129 °C and 122 °C, respectively [39,40,41].

To elucidate further differences in the thermal properties and reveal events that may have been masked due to evaporating solvent, a second heating cycle was performed after cooling the molten samples to room temperature. Figure 4 shows the heating scan of all samples; first heating cycle was initiated from ambient temperature to 170 °C followed by cooling to ambient temperature and a second heating cycle from ambient temperature to 170 °C. As it can be seen in Figure 4, an exothermic peak was formed between 40 °C and 100 °C from the reheated PG–HPMC, PG–SiO_2_ and PG–MCC dispersions, implying recrystallization of amorphous drug. This peak is followed by a melting endotherm with onset at 100–108 °C (Table 3), which corresponds to unstable PG form III. So far, only few studies have reported this peak [42,43].

The impact on the solubility of recrystallization of drug to form III is not clear, e.g., such recrystallization might happen during dissolution of PG–MCC or PG–SiO_2_ dispersions, which recrystallize at low temperatures (Figure 4 and Table 2). The absence of recrystallization peak of form III for PG–HPMCAS (Figure 4) is ascribed to HPMCAS action as nucleation inhibitor. Overall, the DSCs of SE dispersions were eventful, showing presence of forms I and II in the dispersions with HPMCAS, PVP and HPMC but only form I in the dispersions with SiO_2_ and MCC.

#### 3.3.2. DSC of PG Solid Dispersions Prepared by Co-milling

In Figure 5, the DSCs of BM solid dispersions are shown. The melting peaks of dispersions PG–HPMCAS, PG–SiO_2_ and PG–HPMC appear broad implying greater influence on structure integrity. The melting points (Table 4) increased in the order of HPMCAS < SiO_2_ < HPMC < PVP < MCC, which reflects the extent of polymer interference on PG structure. For PG–MCC, the peak at 127.4 °C (Table 4) represents melting of PG form I. The peak of HPMCAS-PG should also correspond to melting of form I despite its appearance at a markedly lower temperature of 98.2 °C (Table 4). This is supported by the fact that in the FTIR spectrum of the co-milled PG–HPMCAS dispersion (Figure 3), the drug exists as form I only. The lowering of m.p. should be attributed to interaction and structure relaxation. On the other hand, the broad peaks of the BM dispersions with HPMC, PVP and SiO_2_ in Figure 5, showing melting points at 115.4, 120.8 and 110.8 °C, respectively (Table 4), should result from the overlapping of melting endotherms of forms I and II, which are present in these dispersions as shown in the FTIR spectra (Figure 3b).

Compared with the SE, the melting points in the DSCs of BM dispersions are lower and the peaks broader (compare Figure 4 and Figure 5) suggesting greater impact on drug structure. One reason is because of the greater amount of polymer present in the solid dispersions (drug/polymer 1:3 in the BM but 3:1 in the SE). A second reason is the high energy supplied by milling, causing local melting and amorphization due to the absorbed heat. The crystallinities of the drug in the SE and BM dispersions compared to neat drug calculated from Equations (1) and (2) are given in Table 3 and Table 4. It can be seen that the crystallinity of PG in the SE dispersions is greater than 96%, whereas in the BM dispersions, it is much lower between 55% and 88%. A third reason is the direct interaction and development of intermolecular forces instead of solvent-mediated interaction.

### 3.4. Solubility Studies

#### 3.4.1. Drug Solubility from Solid Dispersions Prepared by Solvent Evaporation (SE)

The equilibrium solubility of PG from SE solid dispersions and the corresponding physical mixtures is given in Table 5. Except for PG–MCC, all other solid dispersions improved drug solubility compared with the physical mixtures. Among the SE dispersions, the greater increase is shown by PG–HPMCAS (from 17.92 to 23.21 μg/mL) and PG–PVP (from 18.64 to 26.34 μg/mL), which should be attributed to the water solubility of these polymers. During dissolution, soluble units of polymer chains with attached drug molecules may form, facilitating drug wetting and solubility [44]. Furthermore, from Table 5, it can be seen that dispersions with HPMC and SiO_2_ also improved drug solubility, although to a lesser extent than HPMCAS and PVP, which is explained by an initial gel formation due to the penetrating water followed by disintegration to a colloidal dispersion where the drug is homogeneously distributed and wetted [45]. The lack of solubility improvement from the PG–MCC dispersion indicates that despite its hydrophilic character, MCC did not assist drug wetting.

Since solid dispersions produced by SE evolve from dissolved drug and polymer species, their molecules can easily interact during solidification through dispersive, polar and H-bonding forces. This interaction should have a direct effect on drug solubility in water, since greater interaction means stronger attachment of drug to polymer and possible formation of wetted or soluble drug–polymer units. In Figure 6, the relationship between drug solubility in water and drug–polymer Hansen solubility parameter difference Δ*δ*t for dispersions and corresponding physical mixtures is examined. Each pair of points (green and black) refers to each of the studied polymers (data extracted from Table 2 and Table 5). Except for PG–HPMCAS, the points for the solid dispersions fall on a straight-line confirming correlation in four out of five cases. The reason for the PG–HPMCAS point standing as outlier (marked by arrow) might be interference of acetone in the drug–polymer interaction during solvent evaporation. This is supported by the drug solubility vs. Δ*δ*t plot for physical mixtures (Figure 6) where the PG–HPMCAS point lies closer to others.

#### 3.4.2. Drug Solubility from Solid Dispersions Prepared by Co-milling (BM)

The solubilities of PG in water from the BM solid dispersions and the corresponding physical mixtures are presented in Table 6. There is significant improvement over the physical mixtures for HPCAS, HPMC and PVP which is attributed to the polymer solubility in water, facilitating wetting and solubilization of polymer chains attached to drug molecules. The improvement is considerably greater than the corresponding SE dispersions (compare first rows of Table 5 and Table 6), indicating that BM is more effective in promoting intermolecular drug–polymer interactions and structure alteration. In other words, direct drug–polymer interactions operating during BM are more efficient than solvent-mediated interactions. This is supported by the results of thermal analysis for HPCAS, HPMC and PVP in Table 3 and Table 4. BM yielded lower melting points (98.2–120.8 °C compared with 126.1–128.3 °C for form I), lower heats of fusion (51.2–58.8 J/g compared with 73.0–81.9 J/g) and lower crystallinities (46.4%–73.6% compared with 96%–99%) than SE. Further evidence for the greater efficiency of BM is provided for the PG–HPMC dispersion by the FTIR spectrum in Figure 2. The shift due to H-bonding at C20 is minimal for the SE dispersion (from 1707 to 1705 cm^−1^) but pronounced for the BM (from 1707 to 1699 cm^−1^).

On the other hand, from Table 6, it appears that PG–MCC and PG–SiO_2_ solid dispersions prepared by BM had negative effect on drug solubility compared with the physical mixtures. In the case of PG–MCC, this is ascribed to water retention due to the increased MCC amorphous content induced by the milling process, while in the case of PG–SiO_2_, to retention of water by excess silanol groups present on the surface [46,47].

Since interactions during co-milling are restricted to particle surfaces, the extend of interaction should depend on the degree of surface melting, or the temperature increase caused by the frictional forces relative to the polymer glass transition temperature (T_g_), and also on the applied mechanical stress. Assuming that the thermomechanical effect is controlled by the milling conditions, the efficiency of interaction during BM should be controlled both by the similarity of molecular forces expressed as Hansen Δ*δ*t and the T_g_ of the polymer (Table 2). From Table 6, it can be seen that PG combinations that increased its water solubility were made with HPMC, HPMCAS and PVP which have T_g_ less than 162 °C and Δ*δ*t less than 13.8 Mpa^0.5^ (Table 2). On the other hand, combinations with negative impact on drug solubility were with MCC and SiO_2_ which have T_g_ of 143 °C and 250 °C (transition of cristobalite into tetragonal molecular arrangement), and Δ*δ*t of 20.4 and 3.0 Mpa^0.5^, respectively.

From Table 5 and Table 6, it can be seen that for both solid dispersion processes, the solubility enhancement by HPMCAS was greater than HPMC (23.21 μg/mL compared to 16.79 μg/mL for SE and 82.13 μg/mL compared to 56.68 μg/mL for BM) despite their similar chemical structure. The different behaviour of HPMCAS should be attributed to the greater proton donation and hence, H-bonding ability of HPMCAS (Figure 1).

### 3.5. Spectroscopic Analysis of PG–HPMCAS Interactions in the Liquid State

Since PG–HPMCAS dispersion demonstrated exceptional solubility increase (from 34.21 to 82.13 μg/mL, Table 6), its H-bonding performance was further investigated by spectroscopic analysis in the liquid state (i.e., after dissolution of powders). It is possible to understand the mechanism for the enhanced solubility of the PG–HPMCAS by tracing drug–polymer interactions in the liquid state via spectroscopic analysis of dissolved solid dispersions. To fine tune hydrogen bonds between PG and HPMCAS, it is important to mask any hydrogen bonds between water molecules due to the presence of environmental moisture. Because of the strong water absorption in the infrared region, which masks all other peaks, HPMCAS-PG solid dispersions were dissolved in D_2_O instead of H_2_O and the spectra of the solution were then collected. As can be seen in Figure 7, there is a broad stretching vibration at 1735 cm^−1^ due to the C=O group of HPMCAS. Two peaks at 1707 and 1692 cm^−1^ can be seen in the PG spectrum, which belong to the free and bound C20 PG carbonyls, respectively. A significant shift is seen in the spectrum of HPMCAS-PG in methanol at C3 to 1655 and 1635 cm^−1^ (indicated by arrows). As it can be seen in Figure 7 (third spectrum from top), there is significant shift of the large HPMCAS carbonyl peak from 1740 to 1728 cm^−1^, suggesting significant intermolecular interaction with the drug and medium. It is also seen that the two PG peaks at 1699 and 1660 cm^−1^ are shifted to a lower frequency around 1570 cm^−1^, indicating that H-bonding is formed among C=O groups in the PG molecule. The position of the C=O peak at 1570 cm^−1^ did not change at different ratios of H2O/D2O (data not shown), meaning that the H-bonding is formed between the molecules of PG and HPMCAS rather than D2O, in which case, the peak position would have changed with the D2O ratio [48]. Overall, the results of the analysis of the PG–HPMCAS spectra in the liquid state confirm with those of the FTIR on dry dispersions for the excellent H-bonding ability of HPMCAS with PG, which should be associated with the excess protons available for H-bonding with the carbonyl groups of the drug.

### 3.6. Flory–Huggins Thermodynamic Analysis of PG–HPMCAS Interactions

Further confirmation of the miscibility of PG and HPMC is exploited by analysis of the melting endotherms. As can be seen in Figure 8a, significant m.p. depression and heat of fusion reduction of PG–HPMCAS physical mixtures occur as the polymer content increases. The balance between the heat of fusion, melting point and ratio of the drug to the polymer govern the overall mixing tendency within the formed solid dispersion. The thermodynamic treatment showed negative free energy of mixing which was associated with a positive enthalpy (Figure 8b). This trend signifies miscibility. The drop in the enthalpy of mixing required high polymer ratio (90%) (Figure 8b), which reflects the high energy that holds the crystalline structure of PG together. Due to relatively higher T_g_ of PVP and HPMC (150–170 °C), application of this thermodynamic treatment was not possible. Similarly, MCC is partly crystalline while silica exhibits high T_g_ which makes assessment by this approach inappropriate. Therefore, the thermodynamic approach adds further support for the favourable interaction of the drug with HPMCAS.

## 4. Conclusions

The vast number of publications on solid dispersions signifies the unambiguous importance of this promising strategy for improvement of drug solubility. Different methods including state of the art are exploited aiming at effective and reproducible formulations. Among these methods, mechano-chemical method is gaining ground due to the feasibility of application in real industrial production. In this context, it is interesting that the results of the present work proved HPMCAS as the best candidate among other polymers, and co-milling as the better method for the preparation of solid dispersion of PG with highest solubility. Ample evidence for the drug–polymer miscibility was provided by the thermodynamic analysis using both Hansen solubility parameters and Flory–Huggins thermodynamics of mixing, in particular with HPMCAS. Mechano-chemical activation by co-milling has shown great potential to enhance the solubility of the drug despite the fact that the produced particles were not completely amorphous. The results of the present work suggest that prediction of solid miscibility of polymers with the drug can be made using thermal analysis and vibration spectroscopy. The results of the work can be a useful guide in the preparation of PG solid dispersions with minimal batch-to-batch variations and enhanced drug solubility.

## Figures and Tables

**Figure 1 polymers-12-00854-f001:**
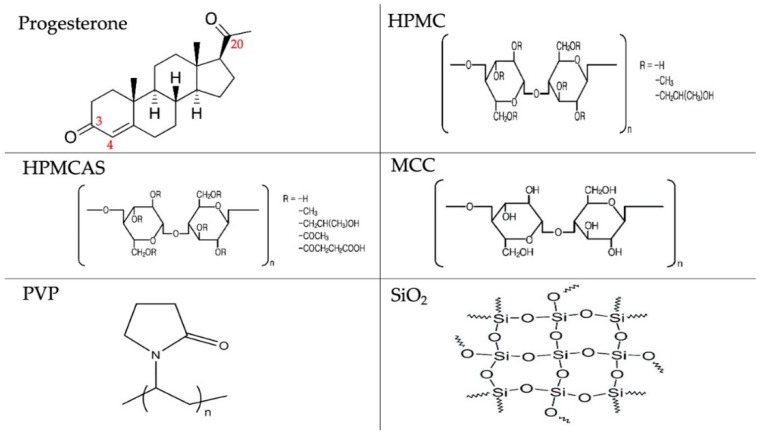
Chemical structures of experimental materials.

**Figure 2 polymers-12-00854-f002:**
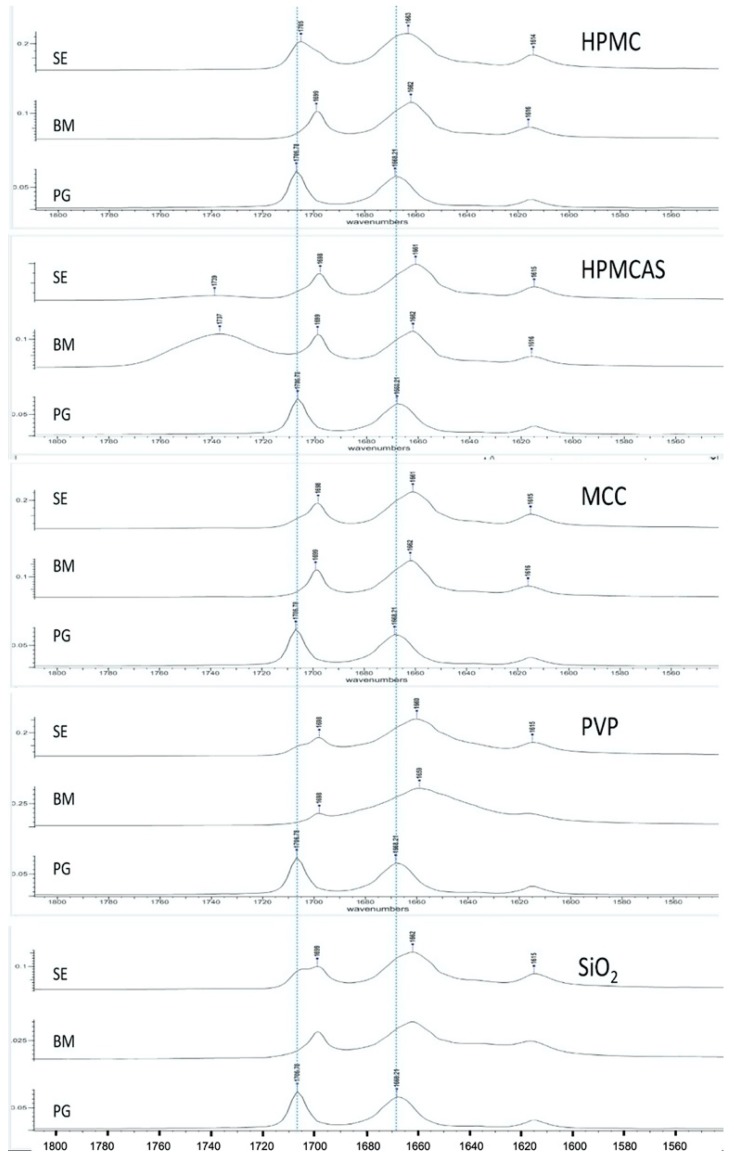
FTIR spectra of PG solid dispersions prepared by solvent evaporation (SE) and bead co-milling (BM) compared with the spectrum of pure drug (progesterone, PG), showing changes in the carbonyl group stretching vibration region (indicated by dotted lines). Peak shifts reflect intermolecular interactions due to the processing method.

**Figure 3 polymers-12-00854-f003:**
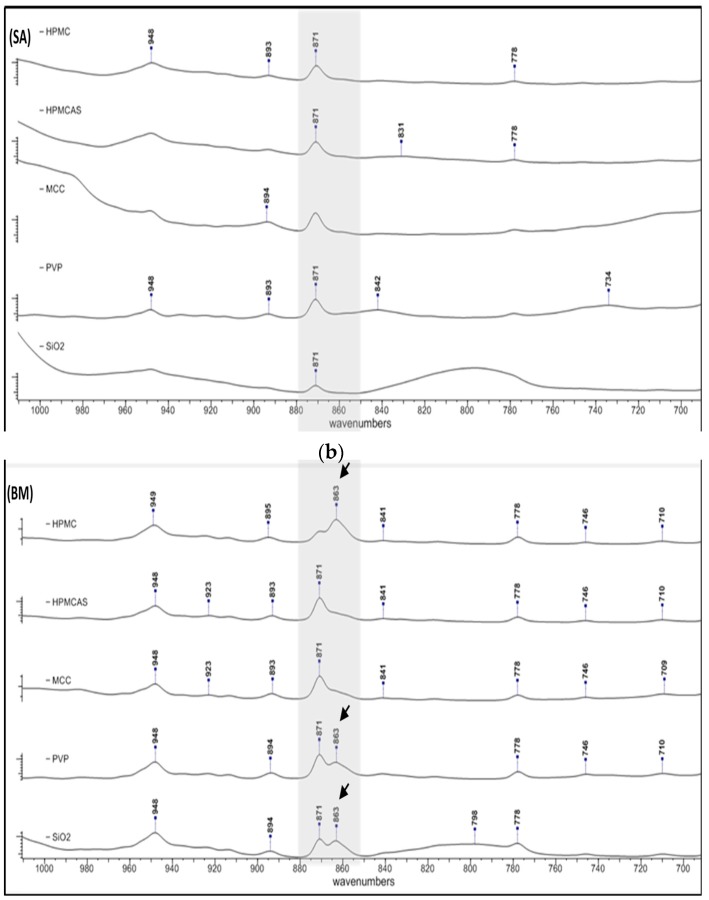
FTIR spectra of solid dispersions prepared by (**a**) solvent evaporation and (**b**) co-milling showing polymorphic changes of PG. In the BM spectra of hydroxypropyl methylcellulose (HPMC), polyvinylpyrrolidone (PVP) and SiO_2_, a second peak appears at 863 cm^−1^ (indicated with arrows) immediately after 871 cm^−1^ (out of plane bending to an *sp^2^* at C2-C4), which is characteristic of PG form II.

**Figure 4 polymers-12-00854-f004:**
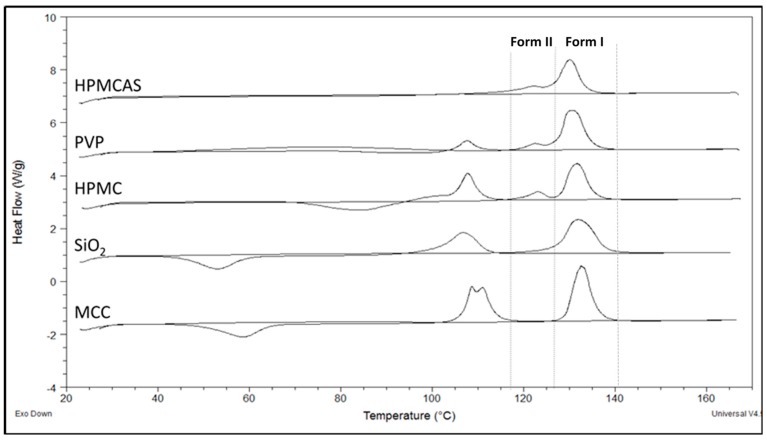
DSC thermograms showing the melting endotherms of PG solid dispersions with the experimental polymers prepared by solvent evaporation.

**Figure 5 polymers-12-00854-f005:**
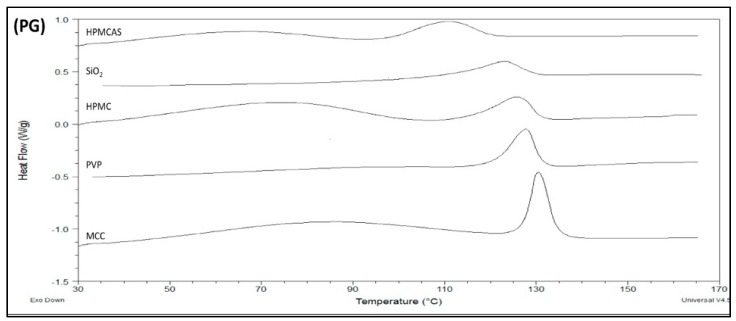
DSC thermograms showing the melting endotherms of PG/polymer solid dispersions prepared by co-milling.

**Figure 6 polymers-12-00854-f006:**
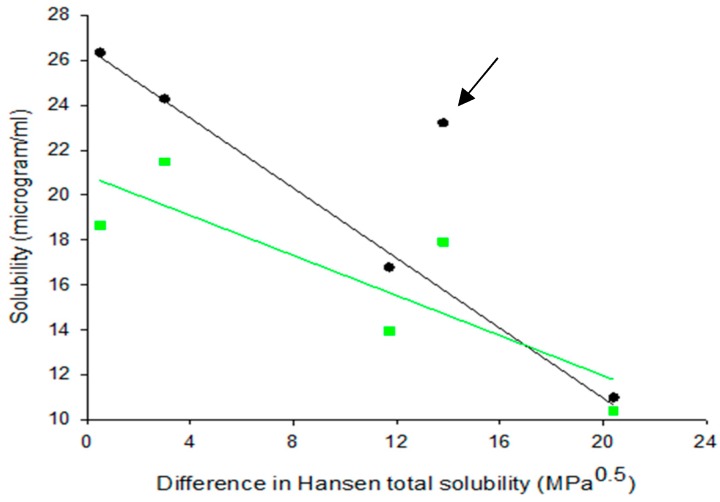
Plots of water solubility of progesterone against drug–polymer Hansen solubility parameter difference for PG–polymer solid dispersions prepared by solvent evaporation (SE) and corresponding physical mixtures (PM) (black symbols correspond to SE and green to BM).

**Figure 7 polymers-12-00854-f007:**
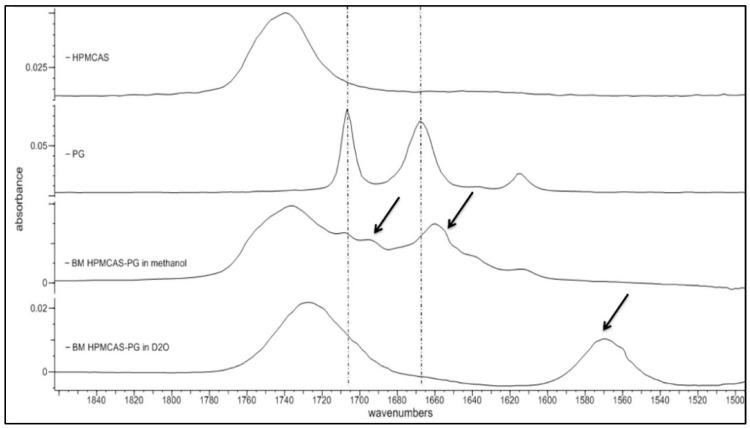
FTIR spectra of PG-hydroxypropyl methylcellulose acetate succinate (PG–HPMCAS) (1:3) solid dispersion prepared by co-milling dissolved in methanol and D_2_O.

**Figure 8 polymers-12-00854-f008:**
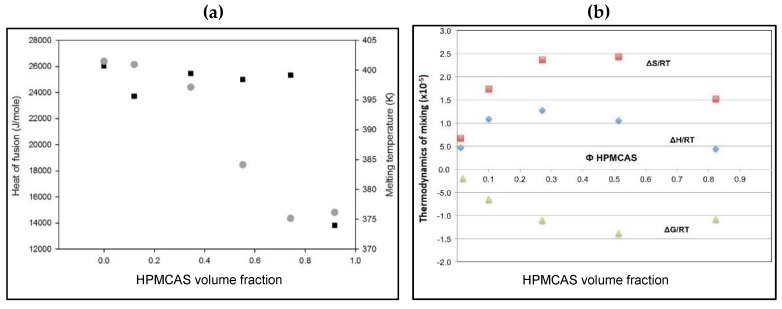
(**a**) Plots of melting point depression (circles) and heat of fusion (squares) of PG in PG–HPMCAS physical mixtures as a function of volume fraction of HPMCAS and (**b**) change of Flory–Huggins thermodynamic parameters: entropy (squares), enthalpy (diamonds) and free energy of mixing (triangles) of progesterone as a function of HPMCAS volume fraction.

**Table 1 polymers-12-00854-t001:** Computation of Hansen solubility parameters (MPa^1/2^) for progesterone and hydroxypropyl methylcellulose acetate succinate (HPMCAS).

**Progesterone**	**Group Contributions (C_i_’s)**			**Occurrences**	***N*_i_*C*_i_**		
First-Order Groups	*δ*_d_’	*δ*_p_’	*δ*_hb_’	*N*i	*δ*_d_’	*δ*_p_’	*δ*_hb_’
CH_3_–	−123.01	−1.6444	−0.7458	3	−369.03	−4.9332	−2.2374
–CH_2_-	1.82	−0.3141	−0.3877	9	16.38	−2.8269	−3.4893
–CH<	82.94	0.6051	−0.2064	3	248.82	1.8153	−0.6192
>C<	182.13	2.0249	−0.0113	2	364.26	4.0498	−0.0226
>C=O	−127.16	0.7691	1.7033	1	−127.16	0.7691	1.7033
–CH=C<	62.48	−1.1018	−1.7171	1	62.48	−1.1018	−1.7171
				*ΣN*i*C*i*:*	*195.75*	−*2.23*	−*6.38*

Second-Order Groups	*δ*_d_’’	*δ*_p_’’	*δ*_hb_’’	*Μ*j	*δ*_d_’’	*δ*_p_’’	*δ*_hb_’’
C_cyclic_=O	−46.57	0.1972	−0.4496	1	−46.57	0.1972	−0.4496
				*ΣMjDj:*	−46.57	0.1972	−0.4496

**HPMCAS**	**Group Contributions (C_i_’s)**			**Occurrences**	***N*_i_*C*_i_**		
First-Order Groups	*δ*_d_’	*δ*_p_’	*δ*_hb_’	(N_i_)	*δ*_d_’	*δ*_p_’	*δ*_hb_’
–CHO-	111.46	1.6001	0.4873	2	222.92	3.2002	0.9746
–CH<	82.94	0.6051	−0.2064	3	248.82	1.8153	−0.6192
–CH_2_–	1.82	−0.3141	−0.3877	1	1.82	−0.3141	−0.3877
–OH	−29.97	1.0587	7.3609	1.5	−44.96	1.5881	11.0414
CH_3_COO–	−53.86	−0.6075	1.7051	0.36	−19.39	−0.2187	0.6138
–CH_2_COO–	89.11	3.4942	1.3893	0.18	16.04	0.6290	0.2501
–CH_2_–	1.82	−0.3141	−0.3877	0.18	0.33	−0.0565	−0.0698
–COOH	−38.16	0.7153	3.8422	0.18	−6.87	0.1288	0.6916
–OCH_3_	−68.07	0.0089	0.2676	0.72	−49.01	0.0064	0.1927
–CH_3_	−123.01	−1.6444	−0.7458	0.24	−29.52	−0.3947	−0.1790
–CH<	82.94	0.6051	−0.2064	0.24	19.91	0.1452	−0.0495
–OH	−29.97	1.0587	7.3609	0.24	−7.19	0.2541	1.7666
–CH_2_O–	13.4	0.8132	−0.1196	0.24	3.22	0.1952	−0.0287
				*ΣNiCi:*	356.11	6.98	14.20

**Table 2 polymers-12-00854-t002:** Hansen solubility parameters (MPa)^1/2^ and glass transition temperatures.

Material	*δ* _d_	*δ* _p_	*δ* _hb_	*δ* _t_	Δ*δ*_t_ (MPa)^1/2^	*T*_g_ (°C)
Progesterone	18.0	5.6	0.9	18.9	–	10
HPMC	18.0	15.3	19.4	30.6	11.7	162
HPMCAS	19.4	14.6	21.9	32.7	13.8	120
MCC	19.4	12.7	31.3	39.3	20.4	143
PVP	17.4	0.6	8.6	19.4	0.5	168
SiO_2_	19.4	7.5	6.7	21.9	3.0	250

**Table 3 polymers-12-00854-t003:** Summary of the results of thermal analysis (DSC) showing melting points and heats of fusion of polymorphs I, II and III, and heat of recrystallization of solid dispersions prepared by solvent evaporation (SE). First heating cycle was initiated from ambient temperature to 170 °C followed by cooling to ambient temperature and a second heating cycle from ambient to 170 °C.

		Polymorph	HPMC	HPMCAS	MCC	PVP	SiO_2_
**First heating cycle**	Melting temp. (onset, °C)	I	128.3 ± 0.2	126.1 ± 1.1	129.1 ± 1.2	127.3 ± 0.7	127.3 ± 0.3
		II	120 ± 0.4	118.8 ± 0.3	N/A	120.1 ± 0.4	N/A
	Heat of fusion (J/g)	I	73 ± 1.4	78.8 ± 1.3	83.3 ± 0.6	81.9 ± 0.5	83.1 ± 0.7
		II	10.4 ± 0.9	2.1 ± 0.2	N/A	2.8 ± 0.2	N/A
**Second heating cycle**	Melting temp. (onset, °C)	III	107.8 ± 0.3	104.6 ± 0.8	106.4 ± 0.2	107.6 ± 0.4	100.4 ± 0.5
	Crystallization temp (onset, °C)	–	69.9 ± 0.7	N/A	49.5 ± 0.4	N/A	53 ± 0.3
	Heat of fusion/crystallization (J/g)	III	26.1 ± 0.8/59.1 ± 1.2	0.7 ± 0.1/NA	63.1 ± 1.1/37.9 ± 1.3	18.4 ± 1.1/NA	57.6 ± 0.1/35.9 ± 0.3

**Table 4 polymers-12-00854-t004:** Melting points of solid dispersions prepared by co-milling (BM). A second cycle was run for SE dispersions because evaporated solvent may have masked some events. There was no issue with the BM dispersions, and for them, a second cycle was not run.

		HPMC	HPMCAS	MCC	PVP	SiO_2_
**First Heating Cycle**	Melting temp. (onset (°C))	115.4 ± 0.2	98.2 ± 0.2	127.4 ± 0.2	120.8 ± 0.3	110.8 ± 0.3
	Heat of fusion (J/g)	52.4 ± 1.4	51.2 ± 1.3	73.6 ± 0.6	58.8 ± 0.5	46.4 ± 0.7
	Crystallinity (%)	62%	61%	88%	70%	55%

**Table 5 polymers-12-00854-t005:** Equilibrium solubility of progesterone (PG) (μg/mL) in water (pH 6.8) from solid dispersions obtained by solvent evaporation (SE), corresponding physical mixtures (PM) and neat drug (PG).

Polymer	HPMC	HPMCAS	MCC	PVP	SiO_2_
SE	16.79 ± 0.25	23.21 ± 0.92	10.99 ± 0.52	26.34 ± 1.64	24.29 ± 5.26
PM	13.95 ± 0.59	17.92 ± 0.58	10.37 ± 0.87	18.64 ± 2.55	21.48 ± 0.57
PG	11.49 ± 0.46				

**Table 6 polymers-12-00854-t006:** Equilibrium solubility (μg/mL) of PG from solid dispersions prepared by co-milling (BM), the corresponding physical mixtures (PM) and neat drug (PG).

Polymer	HPMC	HPMCAS	MCC	PVP	SiO_2_
BM	56.68 ± 2.35	82.13 ± 4.32	14.67 ± 1.31	54.42 ± 5.05	25.18 ± 0.85
PM	42.52 ± 1.53	34.21 ± 1.66	15.67 ± 0.80	50.90 ± 2.45	32.94 ± 0.96
PG	11.49 ± 0.46				

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
