# Peer review of "Solubility Improvement of Progesterone from Solid Dispersions Prepared by Solvent Evaporation and Co-milling"

_polymers, 2020, doi:10.3390/polym12040854_

Round 1

Reviewer 1 Report

In this manuscript, the authors introduce the In this manuscript, the authors introduce the Solubility improvement of progesterone from solid dispersions. It is particularly interesting to improve the solubility of HPMCA-PG solid dispersion ,which might be interesting to many readers.. On the other hand, the manuscript is unclear, the figure description is missing, and the reference needs to be updated. These are issues that need improvement, so please check with the authors again.

However, it still remains some concerns.

1) Introduction: The number of references to the introduction should be added. In addition, the background of research for recent years should be added. (For example, are there any references to the description of line 41-43, and line 48-49?)

2) Line 57-59: PG has low solubility and is easily metabolized. To control metabolic loss, maintaining blood levels requires administration for three days. I need to state in more detail the benefits of improved water solubility.

3) line 111: Number missing.

4) line 129: Is there any significance to the numeric space?

5) Figure : Is the line surrounding the diagram meaningful? If not, you should delete it.

6) Figure : The value on the X axis (wavenumber, etc) is too small to read. Figure does not understand the label description. Should be bigger.

7) Figure 2: Are all PG results the same? Is PG processed by SE or BM? PG description must be provided. In addition, Separate each parameter from PG if the sample is measured. That is easier to understand.

8) Figure 2: It is necessary to explain the gentle peak between 1720 cm-1 and 1760 cm-1 in the BM of HPMCAS.

9) line 229-231: When comparing HPMCAS and MCC, there is no clear difference that HPMC is more noticeable. It is unclear.

10) Figure 3: What do the arrows mean? It is easy to understand if there is an explanation.

11) line 258-260: The polymorphism of PG is shown in FIG. But not shown due to crowdendness. It is unclear and misleading.

12) Figure 4: Is the figure the first heating cycle? Is it the second time? The figure 4 should be explained. The text about figure 4 is also unclear (line 256-258 line272-275).

13) line 274-276: It is unclear. Does it make sense to have arrows only in PVP?

14) line 288: SO2?

15) Table 3 : Table 3 is unclear. This session (3.4.1) will not be rated.

16) Table 6: You should show the table 6 in the session (3.4.2).

17) Figure 7: -HPMCAS has been described as a mixture with PG. Misleading. Should be changed to another description.

18) Figure 8b: X axis should be labeled.

19) I think it is better to observe the solid dispersions with SEM.

20) The references should follow the format.

Author Response

In this manuscript, the authors introduce the Solubility improvement of progesterone from solid dispersions. It is particularly interesting to improve the solubility of HPMCA-PG solid dispersion, which might be interesting to many readers. On the other hand, the manuscript is unclear, the figure description is missing, and the reference needs to be updated. These are issues that need improvement, so please check with the authors again.

RESPONSE TO REVIEWER: We have clarified the manuscript, improved on the presentation and updated the references as much as we could. Our responses are given below in the order received. Changes in the text appear in red colour.

1) Introduction: The number of references to the introduction should be added. In addition, the background of research for recent years should be added. (For example, are there any references to the description of line 41-43, and line 48-49?)

RESPONSE: We added new reference (numbers 5) and we also added more information to the background of research in recent years, lines 41-45, 59-61. Please help us if you have more information, since there is very little recent literature on PG solid dispersions. 

2) Line 57-59: PG has low solubility and is easily metabolized. To control metabolic loss, maintaining blood levels requires administration for three days. I need to state in more detail the benefits of improved water solubility.

RESPONSE: The benefits of improved water solubility have been stated extensively kin lines 65-73.

3) line 111: Number missing.

RESPONSE: section number has been added

4) line 129: Is there any significance to the numeric space?

RESPONSE: Space has been omitted

5) Figure: Is the line surrounding the diagram meaningful? If not, you should delete it.

RESPONSE: Figure was updated, lined deleted

6) Figure: The value on the X axis (wavenumber, etc) is too small to read. Figure does not understand the label description. Should be bigger.

RESPONSE: Both the values on the X axis and label description have been enlarged to make them readable.

7) Figure 2: Are all PG results the same? Is PG processed by SE or BM? PG description must be provided. In addition, Separate each parameter from PG if the sample is measured. That is easier to understand.

RESPONSE: The spectrum labelled ‘PG’ corresponds to pure drug (text has been added in the figure legend). The suggestion to separate each parameter from PG is not clear to us. If the reviewer means spectra, we could do it but the figure will go to the next page. Please help.

8) Figure 2: It is necessary to explain the gentle peak between 1720 cm-1 and 1760 cm-1 in the BM of HPMCAS.

RESPONSE: The following was added to explain the peak “A broad stretching vibration at 1737 cm-1 could be seen due to the C=O group of HPMCAS in the HPMCAS-PG dispersion prepared by BM which slightly shifted to 1739 cm-1 (gentle peak) in HPMCAS-PG dispersion prepared by SE.” (lines 256-258 in the revised).

9) line 229-231: When comparing HPMCAS and MCC, there is no clear difference that HPMC is more noticeable. It is unclear.

RESPONSE:  The sentence “which for HPMC is more noticeable in the BM dispersions” has been removed to avoid ambiguity in analysis.

10) Figure 3: What do the arrows mean? It is easy to understand if there is an explanation.

RESPONSE: The following sentence has been added in the revised (lines 281-283) “In the BM spectra of HPMC, PVP and SiO2 a second peak appears at 863 cm-1 (indicated with arrows) immediately after 871 cm-1 (out of plane bending to an sp2 at C2-C4), which is characteristic of PG form II.” Additionally, the subfigures of Figure 3 are presented enlarged in a cascade fashion so that the details are more readable. 

11) line 258-260: The polymorphism of PG is shown in FIG. But not shown due to crowdendness. It is unclear and misleading.

RESPONSE: Sentence has been removed and new text (lines 314, 315 of revised) has been added to explain why neat drug thermograph is not included.

12) Figure 4: Is the figure the first heating cycle? Is it the second time? The figure 4 should be explained. The text about figure 4 is also unclear (line 256-258 line272-275).

RESPONSE: The following sentence has been added to clarify this part  (revised lines 329-331) ‘Figure 4 shows the heating scan of all samples; 1st heating cycle was initiated from ambient temperature to 170 °C followed by cooling to ambient temperature and a 2nd heating cycle from ambient temperature to 170 °C.  ”

13) line 274-276: It is unclear. Does it make sense to have arrows only in PVP?

RESPONSE: Arrow has been removed.

14) line 288: SO2?

RESPONSE: Corrected as suggested.

15) Table 3 : Table 3 is unclear. This session (3.4.1) will not be rated.

Table 3 was added to summarize the results of thermal analysis. However, if the reviewer considers that the data are already given in the text we could omit it from the submission. The title of Table 3 has been rewritten to clarify its content (lines 322-325 of revised)

16) Table 6: You should show the table 6 in the session (3.4.2).

RESPONSE: Corrected as suggested.

17) Figure 7: -HPMCAS has been described as a mixture with PG. Misleading. Should be changed to another description.

RESPONSE: In the revised manuscript HPMCAS is not described as ‘mixture’ .

18) Figure 8b: X axis should be labeled.

RESPONSE: X axis in Figure 8b has been labelled as the volume fraction of HPMCAS.

19) I think it is better to observe the solid dispersions with SEM.

RESPONSE:  We agree. We plan to include SEM observations in future publications.

20) The references should follow the format.

RESPONSE: We used endnote to format the references (as numbers) and comply with the format of the journal

Reviewer 2 Report

There are several problems with the structure and formatting of the manuscript

The part of manuscript named "results" should be changed for "Results and discussion". In this part there are references to publications-means discussion with other results so it can not be only "Results"

It shoud be specified whether Tables 1 and 2 are literature data or test / calculation results. If they are from the literature they should be citated, and if there are the results, they should be in the results part of manuscript, not in materials and methods.

Figure 1 it should be placed rather in introduction, not in results...

The resolution of the figures is not good and should be improved.

Tables 5 and 6 are are badly formatted.

Abreviation "BE" is used two times and not explained. It shoud be "BM" instead?

Format of "Introduction" is aligned, the rest of the manuscript is not. Please standardize. 

Fourier transform infrared studies (FTIR) - "studies" is not well used. It should be rather "Fourier transform infrared spectroscopy (FTIR)"

Numerous, among others, in the listed formulas should be in the subscript (-OCH2COCH3) (-OCOCH2CH2COOH) (-OCH3) (OCH2CH(OH)CH3), page 4

Author Response

There are several problems with the structure and formatting of the manuscript

RESPONSE: The authors would like to thank the reviewer for the insightful feedback.  We have clarified the manuscript and revised the structure and headings.

The part of manuscript named "results" should be changed for "Results and discussion". In this part there are references to publications-means discussion with other results so it can not be only "Results"

RESPONSE: Corrected as suggested

It shoud be specified whether Tables 1 and 2 are literature data or test / calculation results. If they are from the literature they should be citated, and if there are the results, they should be in the results part of manuscript, not in materials and methods.

RESPONSE: The requested information is provided as footnote of Table 2 (lines 232-237)

Figure 1 it should be placed rather in introduction, not in results...

RESPONSE: Figure 1 has been moved to the introduction.

The resolution of the figures is not good and should be improved.

RESPONSE: We tried to improve the resolution of the Figures.

Tables 5 and 6 are  badly formatted.

RESPONSE: Format of Tables 5, 6 has been revised.

Abreviation "BE" is used two times and not explained. It shoud be "BM" instead?

RESPONSE: We would like to thank the reviewer for identifying those typos, we have corrected them to BM

Format of "Introduction" is aligned, the rest of the manuscript is not. Please standardize. 

RESPONSE: Corrected as suggested

Fourier transform infrared studies (FTIR) - "studies" is not well used. It should be rather "Fourier transform infrared spectroscopy (FTIR)"

RESPONSE: Corrected as suggested by the reviewer (line 129)

Numerous, among others, in the listed formulas should be in the subscript (-OCH2COCH3) (-OCOCH2CH2COOH) (-OCH3) (OCH2CH(OH)CH3), page 4

RESPONSE: They have been corrected (lines 170, 171)

Reviewer 3 Report

The manuscript presented the methods to improve the solubility of progesterone, including solvent evaporation and co-milling, and the effects of polymers on the solubility of progesterone were studied. The results showed the potential of these two methods for the development and utilization of several poorly soluble drugs. Therefore, I recommend the paper can be published after minor revision.

  1. There are many grammatical errors throughout the paper, and the font format is not uniform, for example, in line76 ‘’Mw’’ should be written ’’MW’’. The English need to be polished.
  2. In line 90, in front of the ‘’Preparation of PG solid dispersions using mechanochemical activation’’ this sentence should be numbered, ‘’2.2.2 Preparation of PG solid dispersions using mechanochemical activation‘’. The same error occurred in line 111, and the subsequent numbers need to be adjusted.
  3. Chemical formula should be standardized, for example in line 147, -CH< should be -CH=, in line 152’’ (-OCH2COCH3) ’’ should be (-OCH2COCH3).
  4. Table should be standardized, for example Table 5, and in line 339 ’’μg/mlshould be ’’μg/mL’’.
  5. In Figure 6, the lines of two different colors each represent what should be identified.
  6. In line 169, ’’ TMpure’’ should be consistent with the expression in line 168.

Author Response

The manuscript presented the methods to improve the solubility of progesterone, including solvent evaporation and co-milling, and the effects of polymers on the solubility of progesterone were studied. The results showed the potential of these two methods for the development and utilization of several poorly soluble drugs. Therefore, I recommend the paper can be published after minor revision.

 RESPONSE: The authors would like to thank the reviewer for the comments.

  1. There are many grammatical errors throughout the paper, and the font format is not uniform, for example, in line76 ‘’Mw’’ should be written ’’MW’’. The English need to be polished.

RESPONSE: Mw has been changed to MW (line 94). Further checks were made and grammatical mistakes were corrected.

In line 90, in front of the ‘’Preparation of PG solid dispersions using mechanochemical activation’’ this sentence should be numbered, ‘’2.2.2 Preparation of PG solid dispersions using mechanochemical activation‘’. The same error occurred in line 111, and the subsequent numbers need to be adjusted.

RESPONSE: It has been corrected (line 108).

  1. Chemical formula should be standardized, for example in line 147, -CH< should be -CH=, in line 152’’ (-OCH2COCH3) ’’ should be (-OCH2COCH3).

RESPONSE: We have made the requested change (lines 165, 170, 171)

  1. Table should be standardized, for example Table 5, and in line 339 ’’μg/ml’ should be ’’μg/mL’’.

RESPONSE: It is now consistently reported as μg/mL throughout the manuscript.

  1. In Figure 6, the lines of two different colors each represent what should be identified.

RESPONSE: The legend has been clarified to explain that  (black symbols correspond to SE, green to PM). 

  1. In line 169, ’’ TMpure’’ should be consistent with the expression in line 168.

RESPONSE: Correction has been made (line 187)

Round 2

Reviewer 1 Report

 In this manuscript, the authors introduce the In this manuscript, the authors introduce the Solubility improvement of progesterone from solid dispersions. It is particularly interesting to improve the solubility of HPMCA-PG solid dispersion ,which might be interesting to many readers. These studys may help improve dissolution of other drugs. The work is well performed and the results are sound.